# AR-RAG: Autoregressive Retrieval Augmentation for Image Generation

**Jingyuan Qi**[* 1]   **Zhiyang Xu**[* 1]   **Qifan Wang**[2]   **Lifu Huang**[3]

[1]Virginia Tech   [2]Meta   [3] UC Davis

jingyq1@vt.edu   lfuhuang@ucdavis.edu

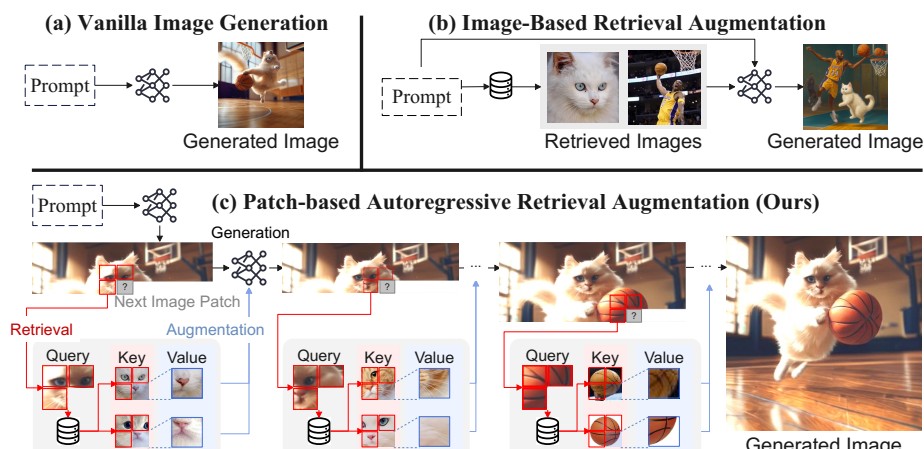

Figure 1: Comparison between Autoregressive Retrieval Augmentation (AR-RAG) for image generation in (c) and existing image generation paradigms in (a) (b). In AR-RAG, image patches in red boxes denote retrieval queries and keys, image patches in blue boxes are retrieved values, and gray boxes with the question mark are next image patches to be predicted. (Caption: *A white cat is playing basketball on the court.*)

## Abstract

We introduce Autoregressive Retrieval Augmentation (AR-RAG), a novel paradigm that enhances image generation by autoregressively incorporating k-nearest neighbor retrievals at the patch level. Unlike prior methods that perform a single, static retrieval before generation and condition the entire generation on fixed reference images, AR-RAG performs context-aware retrievals at each generation step, using prior-generated patches as queries to retrieve and incorporate the most relevant patch-level visual references, enabling the model to respond to evolving generation needs while avoiding limitations (e.g., over-copying, stylistic bias, etc.) prevalent in existing methods. To realize AR-RAG, we propose two parallel frameworks: (1) Distribution-Augmentation in Decoding (DAiD), a training-free plug-and-use decoding strategy that directly merges the distribution of model-predicted patches with the distribution of retrieved patches, and (2) Feature-Augmentation in Decoding (FAiD), a parameter-efficient fine-tuning method that progressively smooths the features of retrieved patches via multi-scale convolution operations and leverages them to augment the image generation process. We validate the effectiveness of AR-RAG on widely adopted benchmarks, including Midjourney-30K, GenEval, RareBench, T2I-Bench, and DPG-Bench,

---

[1]Jingyuan Qi and Zhiyang Xu contributed equally to this work.

39th Conference on Neural Information Processing Systems (NeurIPS 2025).

demonstrating significant performance gains over state-of-the-art image generation models.[1]

# 1  Introduction

Recent advancements in image generation have demonstrated remarkable capabilities in producing photorealistic images based on user prompts [33, 30, 7, 39, 10, 43, 9, 45, 47, 28, 6]. However, despite these improvements, the generated images often exhibit local distortions and inconsistencies, particularly in visual objects that possess complex structures [11], frequently interact with other objects and the surrounding scene [23, 27], or are underrepresented in the training data [8]. A promising approach to mitigating these challenges is retrieval-augmented generation (RAG), which enhances the generation process by incorporating real-world images as additional references [8, 3]. While RAG has been extensively explored in the language domain [24, 13], its application to image and multimodal generation remains largely underdeveloped. A few existing studies [3, 8, 48, 50, 51] bridge this gap by performing a single-step retrieval based on the input prompt prior to generation, conditioning the entire image generation process on fixed visual cues (Figure 1 (b)). However, as demonstrated in our pilot study (Section 5.2), such static, coarse-grained retrieval approaches [3, 8, 51] frequently introduce irrelevant or weakly aligned visual contents that persist throughout generation. Since the retrieved images are selected once, before decoding begins, and remain unchanged, these methods cannot respond to the evolving generation needs, resulting in over-copying of irrelevant details, stylistic bias, and the hallucination of unrelated visual elements. For example, as shown in Figure 1(b), a basketball player present in the retrieved references, despite being irrelevant to the input prompt, unintentionally appears in the generated image.

In this paper, we propose Autoregressive Retrieval Augmentation (AR-RAG), a novel retrieval-augmented paradigm for image generation that dynamically and autoregressively incorporates patch-level k-nearest-neighbor (k-NN) retrievals throughout the generation process (Figure 1(c)). In contrast to prior methods that rely on static, coarse-grained retrievals of entire reference images, typically using captions as retrieval queries and keys, AR-RAG performs fine-grained, step-wise retrieval at the image patch level. Specifically, as generation unfolds, AR-RAG leverages the already-generated surrounding patches as localized queries to retrieve contextually similar patches from a pre-constructed patch-level database. This database is built by encoding real-world images into latent patch features, where each entry contains a patch embedding as a value and the embeddings of its $h$-hop spatial neighbors as a key. During the generation of the next target patch (gray boxes in Figure 1(c)), AR-RAG retrieves the top-$K$ most relevant patches (blue boxes) by measuring similarity between the surrounding generated context patches (red boxes) and database keys (also red boxes). These retrieved patches are then integrated into the model to inform and enhance the prediction of the next patch, enabling the model to dynamically adjust to local generation needs. By conditioning on the evolving generation context as retrieval queries, AR-RAG ensures that retrieved visual references remain relevant throughout the generation process, encouraging local semantic coherence. Moreover, the patch-level retrieval allows for precise integration of visual elements without overcommitting to entire reference images, avoiding the limitations of over-copying or irrelevant conditioning observed in static retrieval.

To realize the AR-RAG framework, we introduce two parallel implementations: (1) **Distribution-Augmentation in Decoding (DAiD)**, a training-free, plug-and-play decoding strategy that merges the model's predicted patch distribution with that of the retrieved patches. Specifically, the top-$K$ retrieved patches are assigned probabilities inversely proportional to their normalized $\ell_2$ distances computed from the query and key patch embeddings. These probabilities are then linearly combined with the model's native output distribution to guide the next patch prediction, enabling retrieval-aware generation without any additional training. (2) **Feature-Augmentation in Decoding (FAiD)**, a parameter-efficient fine-tuning approach that integrates retrieved patches into the generation process through learned smoothing and blending mechanisms. Specifically, when generating the next image token, FAiD operates in two stages: (1) refining the retrieved patch features by adjusting them to better fit the local context of the already generated surrounding patches, based on parameterized convolutional operations of varying kernel sizes; and (2) blending the refined features of retrieved patches with the model's predicted feature representation for the next patch, based on compatibility

---

[1]Code and model checkpoints can be found at `https://github.com/PLUM-Lab/AR-RAG`.

scores computed for each retrieved patch to quantify their alignment with the current generation context. To enable iterative refinement, we insert multiple FAiD modules at selected transformer layers, where the output of each FAiD module, i.e., the context-aware retrieved features blended at that layer, is forwarded as input to the next FAiD module in deeper layers. This progressive retrieval refinement mechanism allows the model to incrementally enhance its predictions as patch-level representations evolve through the network. We evaluate AR-RAG on five widely adopted benchmarks, including Midjourney-30K [2], Geneval [14], RareBench [29], T2I-Bench [19], and DPG-Bench [18]. Experimental results demonstrate that both DAiD and FAiD significantly improve the coherence and naturalness of generated images while introducing only marginal computational overhead.

The contributions of our work can be summarized as follows:

- We propose AR-RAG, the first patch-level autoregressive retrieval augmentation framework which dynamically retrieves and integrates fine-grained visual content to enhance image generation, while avoiding limitations (e.g., over-copying, stylistic bias, etc.) prevalent in existing image-level retrieval augmentation methods.

- We introduce Distribution-Augmentation in Decoding (DAiD), a training-free, plug-and-play decoding strategy that directly integrates the distribution of retrieved patches into that predicted by the image generation models, enabling easy integration into existing architectures.

- We introduce Feature-Augmentation in Decoding (FAiD), a parameter-efficient fine-tuning framework that progressively refines and blends retrieval signals via lightweight convolutional modules, enhancing spatial coherence and visual quality across layers.

- Extensive experiments and analysis show that AR-RAG significantly improves performance of state-of-the-art image generation model across diverse metrics. In particular, Janus-Pro with FAiD achieves 6.67 FID on Midjourney-30K and 0.78 overall score on GenEval, establishing a new state of the art among autoregressive image generation models of comparable scale.

## 2 Preliminary

**Autoregressive Image Generation Models** We implement both DAiD and FAiD based on Janus-Pro [9], an autoregressive (AR) unified generation model, due to its strong performance. Janus-Pro is initialized from a transformer-based pre-trained large-language model [2], and employs a quantized autoencoder [39] to encode images into discrete image tokens. During multimodal pretraining, the model learns to predict a sequence of discrete image tokens $[v_1, v_2, ...v_N]$ conditioned on an input text prompt $[t_1, t_2, ...t_M]$. The training objective is formally defined as:

$$\arg\max_{\phi} \sum_{}^{\mathcal{D}} \sum_{n=1}^{N} P_{\phi}(v_n|t_1, t_2, ..., t_M, v_1, ...v_{n-1}) \tag{1}$$

where $\mathcal{D}$ is the training corpus. This is the same training objective used in our FAiD method in Section 3.3. We argue that DAiD and FAiD can be extended to any image generation model that autoregressively predicts probability distributions of discrete image tokens such as LlamaGen [39], Show-o [46] and VAR [40].

**Quantized Autoencoder** The quantized autoencoder used in Janus-Pro consists of an encoder $\theta_{\text{enc}}$, a decoder $\theta_{\text{dec}}$, and a codebook $\mathcal{Z}$. The encoder, a convolutional neural network, downsamples and compresses raw pixel inputs into compact patch representations. During the quantization process, each patch representation is mapped to an index in the codebook by identifying its nearest neighbor vector in the codebook. In the decoding stage, these patch indices are mapped back to their corresponding vector representations via the codebook, and the decoder, another convolutional neural network, reconstructs the image from these compact representations. In our implementation, we leverage this autoencoder to build the coupled database for Janus-pro which is detailed in Section 3.1.

---

[2] `https://huggingface.co/datasets/playgroundai/MJHQ-30K`

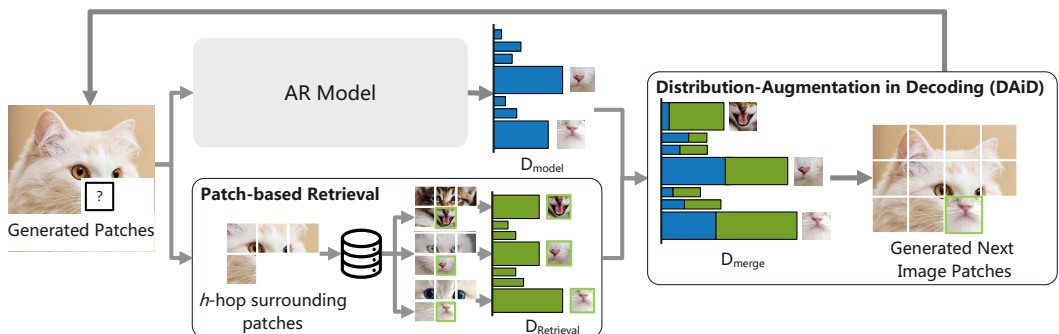

Figure 2: The decoding process in Distribution-Augmentation in Decoding (DAiD).

# 3 AR-RAG: Patch-based Autoregressive Retrieval Augmentation

## 3.1 Patch-based Retrieval Database Construction

We build a patch-based retrieval database based on several large-scale, real-world image datasets, including CC12M [5] and JourneyDB [38]. Specifically, for each image $I$, we encode it into $N$ patches using the quantized autoencoder [39], $\theta_{\text{Enc}}$, from Janus-Pro: $\mathbf{V} = \theta_{\text{enc}}(I) \in \mathbb{R}^{\sqrt{N} \times \sqrt{N} \times d}$, where $d$ is the hidden dimension, and $\mathbf{V}_{ij}$ corresponds to the latent representation of the patch at position $(i, j)$. We utilize each patch vector $\mathbf{V}_{ij}$ as the value of a database entry and the representation of its $h$-hop surrounding patches as the key. Here, the $h$-hop surrounding patch representation is formed by concatenating the vectors of adjacent patches centering around $(i, j)$ in a top-to-bottom, left-to-right order. For example, for a patch at position $(i, j)$, the 1-hop surrounding representation spans 8 surrounding patches $[\mathbf{V}_{(i-1)(j-1)} : \mathbf{V}_{(i-1)(j)} : \mathbf{V}_{(i-1)(j+1)} : \mathbf{V}_{(i)(j-1)} : \mathbf{V}_{(i)(j+1)} : \mathbf{V}_{(i+1)(j-1)} : \mathbf{V}_{(i+1)(j)} : \mathbf{V}_{(i+1)(j+1)}]$ where : denotes the concatenation operation of image patch features. If a patch is located at the edge of the image and lacks certain surrounding patches, we substitute each missing surrounding patch with a zero vector $\mathbf{0}$.

## 3.2 Distribution-Augmentation in Decoding (DAiD)

Given a text prompt $T$, Janus-Pro autoregressively predicts a sequence of image tokens $[v_1, v_2, ...v_N]$ where per-token probability is defined in Equation 1. As shown in Figure 2, DAiD augments this process by incorporating probability distributions from retrieved image patches. Specifically, when Janus-Pro predicts the next image token $v_{ij}$, we first utilize the codebook $\mathcal{Z}$ to convert $v_{ij}$'s $h$-hop already generated surrounding patches into patch representations. If no surrounding image tokens are available at a given position (e.g., when $i = 0$ or $j = 0$), we use the zero vector $\mathbf{0}$ as a placeholder. Once we compute the representation of $v_{ij}$'s $h$-hop surrounding patches, we leverage it as the retrieval query and retrieve the top-$K$ most similar patch representations from the database constructed in Section 3.1 using $l_2$ distance. We denote the representations of the top-$K$ retrieved patches as $[\hat{\mathbf{v}}_1, \hat{\mathbf{v}}_2, ..., \hat{\mathbf{v}}_K]$ and their corresponding $l_2$ distances as $[s_1, s_2, ..., s_K]$. These retrieved representations are then mapped back to discrete token indices using the codebook: $\hat{v}_k = \mathcal{Z}(\hat{\mathbf{v}}_k)$.

To augment the generation process with the retrieved image tokens $[\hat{v}_1, \hat{v}_2, ..., \hat{v}_K]$, we create a retrieval-based distribution $D_{\text{retrieval}} \in \mathbb{R}^{|\mathcal{Z}|}$ over the entire codebook $\mathcal{Z}$, where $|\mathcal{Z}|$ is the codebook size. Tokens not included in the top-$K$ retrieved set are assigned a probability of 0. For tokens within the top-$K$, we compute their probabilities using a softmax over their $l_2$ distance to the query, scaled by a retrieval temperature hyperparameter $\tau$:

$$D_{\text{retrieval}}[v] = \begin{cases} p(\hat{v}_k) & \text{if } v = \hat{v}_k \text{ for some } m \in \{1, 2, ..., K\} \\ 0 & \text{otherwise} \end{cases} \tag{2}$$

$$p(\hat{v}_k) = \frac{\exp(-s_k/\tau)}{\sum_{k=1}^{K} \exp(-s_k/\tau)}, \tag{3}$$

This creates a sparse distribution where only the top-$K$ retrieved tokens have non-zero probabilities. Finally, we merge this retrieval distribution with the model's predicted distribution $D_{\text{model}}$ using a

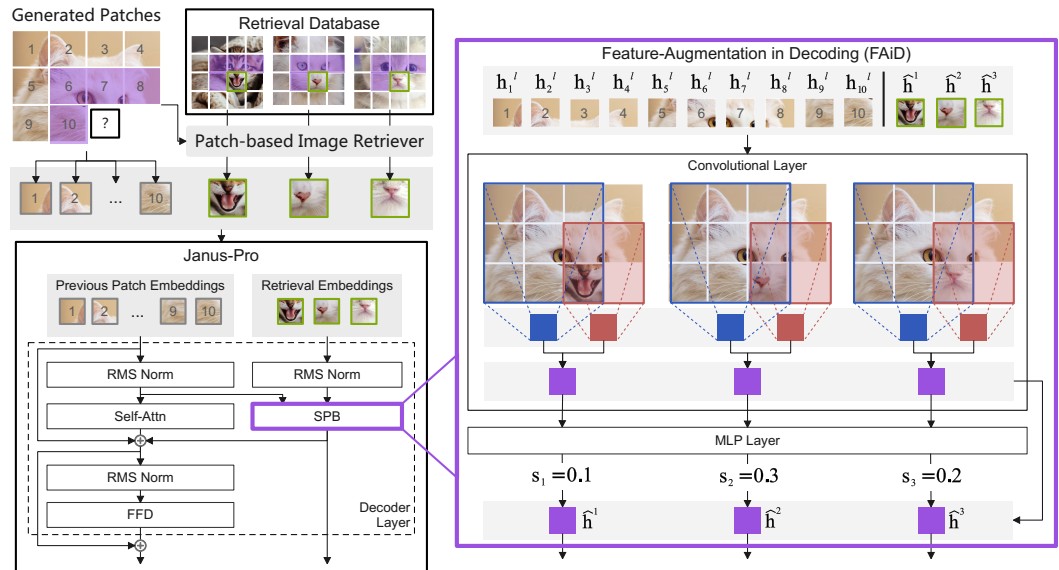

Figure 3: Overall architecture of Feature-Augmentation in Decoding (FAiD).

weighted average:

$$D_{\text{merge}} = (1 - \lambda) \cdot D_{\text{model}} + \lambda \cdot D_{\text{retrieval}}, \tag{4}$$

where $\lambda \in [0, 1]$ is the retrieval weight hyperparameter controlling the influence of retrieved patches on the final distribution. The next token is then sampled from this merged distribution: $v_{ij} \sim D_{\text{merge}}$.

### 3.3 Feature-Augmentation in Decoding (FAiD)

While DAiD offers a training free approach to directly augment the probability distribution of predicted patches using retrieved ones, it suffers from noise propagation and limited flexibility in fully leveraging the fine-grained visual information in the retrieved patches. We thus further propose FAiD, a feature-based autoregressive augmentation strategy to enhance the image generation process. As illustrated in Figure 3, when predicting the next token $v_{ij}$ during image generation, we employ the same retrieval process described in Section 3.2 to obtain the top-$K$ most relevant patches and their representations $[\hat{\mathbf{v}}_1, \hat{\mathbf{v}}_2, ..., \hat{\mathbf{v}}_K]$ from our database. To effectively incorporate them into the autoregressive generation process, FAiD consists of two steps: (1) refining retrieved patches to ensure coherence with the surrounding context of $v_{ij}$ in the generated image, and (2) adaptively blending the representation of refined patches with the hidden state of the predicted next patch based on learned compatibility scores. To enable progressive refinement of retrieved information as representations evolve through the network, we insert a FAiD module for every $L/b$ decoder layers of the generation model, where $L$ denotes the total number of decoder layers and $b$ is a hyperparameter.

**Multi-Scale Feature Smoothing** The key of effective patch integration lies in ensuring spatial coherence between retrieved patches and the surrounding image context. To achieve this, we propose *multi-scale feature smoothing* (Algorithm 1 in Appendix A), where multi-scale convolutions are applied to retrieved patches within the generation context, so that the retrieved visual features are smoothed to preserve structural and stylistic consistency with the surrounding context of the predicted token. Specifically, at each step when predicting the next image token $v_{ij}$, we first construct a 2D spatial representation $\mathbf{H}^l \in \mathbb{R}^{\sqrt{N} \times \sqrt{N} \times D}$ of the current partially-generated image by arranging their hidden states $[\mathbf{h}_1^l, \mathbf{h}_2^l, ..., \mathbf{h}_{n-1}^l, \mathbf{h}_n^l]$ from the current decoder layer $l$. We use $\mathbf{0}$ vectors as placeholders for positions that have not yet been generated. Then, we transform the retrieved patch representations $[\hat{\mathbf{v}}_1, \hat{\mathbf{v}}_2, ..., \hat{\mathbf{v}}_K]$ into the generation model's hidden space by mapping each patch $\hat{\mathbf{v}}_k$ to a discrete token index via the codebook $\mathcal{Z}$ and embedding it through the pretrained image embedding layer $\text{Emb}_{\text{img}}$:

$$[\hat{\mathbf{h}}_1, \hat{\mathbf{h}}_2, ..., \hat{\mathbf{h}}_K] = \text{Emb}_{\text{img}}([\mathcal{Z}(\hat{\mathbf{v}}_1), \mathcal{Z}(\hat{\mathbf{v}}_2), ..., \mathcal{Z}(\hat{\mathbf{v}}_K)]) \tag{5}$$

For each retrieved patch $\hat{\mathbf{h}}_k$, we create a copy of $\mathbf{H}^l$ where position $(i, j)$ (the location of $v_{ij}$) is replaced with $\hat{\mathbf{h}}_k$. We then apply convolution operations at multiple scales ($2 \times 2$ through $Q \times Q$) to capture contextual patterns at different resolutions. To maintain computational efficiency, we only perform convolution operations when the kernel covers position $(i, j)$, rather than processing the entire image. Each convolution kernel $\text{Conv}_{q \times q}$ produces a refined representation $\hat{\mathbf{h}}_k^q$ for the retrieved patch at scale $q$. The final refined representation for each retrieved patch is computed as a weighted sum of these multi-scale features:

$$\hat{\mathbf{h}}_k \leftarrow \sum_{q=2}^{Q} \text{softmax}(\mathbf{\Omega})_q \cdot \hat{\mathbf{h}}_k^q \tag{6}$$

where $\mathbf{\Omega} = [\omega_2, ..., \omega_Q]$ are learnable parameters that determine the importance of each scale.

**Feature Augmentation**   After feature smoothing, some of the retrieved patch features may still not be able to fit into the surrounding neighbors and hence we need to lower their impact in the final representation. Thus, we compute a compatibility score for each of the refined patches. This is achieved by projecting each refined retrieved patch representation through a linear transformation parameterized by a weight matrix $\mathbf{W} \in \mathbb{R}^{1 \times D}$, yielding the score $s_k = \hat{\mathbf{h}}_k \mathbf{W}^T$. The final representation for the next image token $v_{ij}$ after layer $j$ is computed as:

$$h_{ij}^{(l+1)} = h_{ij}^l + \Delta h_{ij}^l + \sum_{k=1}^{K} s_k \hat{\mathbf{h}}_k \tag{7}$$

Here, $h_{ij}^l$ is the residual, $\Delta h_{ij}^l$ is the updated representation from the transformer layer $l$, and $\sum_{k=1}^{K} s_k \hat{\mathbf{h}}_k$ is the contribution of the retrieved image patches.

# 4   Experiment Setup

**Patch-based Retrieval Database**   To construct our patch-level retrieval database, we randomly sample 5.7 million images from CC12M [5], 3.3 million from JourneyDB [38], and 4.6 million from DataComp [12], while ensuring that any samples included in the testing set are excluded to prevent data leakage. Each image is encoded into a sequence of patch-level representations and image tokens using the same image tokenizer employed in the Janus-Pro model. For efficient similarity search, we implement our retriever using the FAISS library [22].

**Training Setup**   We adopt Janus-Pro-1B [9] and Show-o [46] as our backbone models and fine-tune them on a dataset of 50,000 image-caption pairs sampled from CC12M [5] and Midjourney-v6 [3]. We empirically determine the optimal hyperparameters for DAiD and FAiD, and the complete hyperparameter optimization experiment results can be found in Appendix C.2. Further details regarding the training dataset construction and implementation can be found in Appendix B.3.

**Baselines**   To evaluate the effectiveness of our proposed methods, we adopt several state-of-the-art image generation approaches as baselines, including non-retrieval models such as LlamaGen [39], LDM [34], Stable Diffusion (SDv1.5 and SDv3) [33, 10], PixArt-alpha [7], DALL-E 2 [32], Show-o [46], Janus-Pro [9], and FLUX.1-dev [4], and image-based retrieval augmentation methods, including RDM [3], RA-CM3 [48], and ImageRAG[35]. Since pretrained models of RA-CM3 are not publicly available, we try our best to replicate their method based on Janus-Pro to ensure a fair comparison. More details of training and implementation of RA-CM3 can be found in Appendix B.1.

**Evaluation Benchmarks and Metrics**   To comprehensively evaluate our proposed methods, we employ five benchmarks: (1) GenEval [14], which assesses models' ability to generate images with specific attributes and relationships described in text prompts; (2) DPG-Bench [18], which evaluates performance on detailed prompts with complex requirements; (3) RareBench [29], which is designed to evaluate the ability of diffusion models to generate rare compositions of concepts; (4) T2I-Bench [19], which focuses on fine-grained attribute binding across four key dimensions: color, shape, texture, and numeracy; and (5) Midjourney-30k [42], where we employ three complementary metrics: FID [17] for measuring statistical similarity between generated and real image distributions, CMMD [21] for assessing alignment with human perception using CLIP embeddings, and FWD [41]

---

[3] https://huggingface.co/datasets/brivangl/midjourney-v6-llava
[4] https://huggingface.co/black-forest-labs/FLUX.1-dev

| Method | Params | Single Obj. | Two Obj. | Counting | Colors | Position | Color Attri. | Overall ↑ |
|---|---|---|---|---|---|---|---|---|
| *Non Retrieval-Augmented Model* | | | | | | | | |
| PixArt-$\alpha$ | 0.6B | 0.98 | 0.50 | 0.44 | 0.80 | 0.08 | 0.07 | 0.48 |
| LlamaGen | 0.8B | 0.71 | 0.34 | 0.21 | 0.58 | 0.07 | 0.04 | 0.32 |
| SDv1.5 | 0.9B | 0.97 | 0.38 | 0.35 | 0.76 | 0.04 | 0.06 | 0.43 |
| SDv2.1 | 0.9B | 0.98 | 0.51 | 0.44 | 0.85 | 0.07 | 0.17 | 0.50 |
| Janus-Pro | 1.0B | 0.98 | 0.77 | 0.52 | 0.84 | 0.61 | 0.55 | 0.71 |
| Show-o | 1.3B | 0.98 | 0.80 | **0.66** | 0.84 | 0.31 | 0.50 | 0.68 |
| LDM | 1.4B | 0.92 | 0.29 | 0.23 | 0.7 | 0.02 | 0.05 | 0.37 |
| SD3 (d=24) | 2.0B | 0.98 | 0.74 | 0.63 | 0.67 | 0.34 | 0.36 | 0.62 |
| SDXL | 2.6B | 0.98 | 0.74 | 0.39 | 0.85 | 0.15 | 0.23 | 0.55 |
| DALL-E 2 | 6.5B | 0.94 | 0.66 | 0.49 | 0.77 | 0.10 | 0.19 | 0.52 |
| DALL-E 3 | - | 0.96 | 0.87 | 0.47 | 0.83 | 0.43 | 0.45 | 0.67 |
| Transfusion | 7.3B | - | - | - | - | - | - | 0.63 |
| Chameleon | 34B | - | - | - | - | - | - | 0.39 |
| FLUX.1-dev | 12B | - | - | - | - | - | - | 0.67 |
| *Retrieval-Augmented Model* | | | | | | | | |
| RDM | 1.4B | 0.91 | 0.21 | 0.28 | 0.71 | 0.02 | 0.04 | 0.36 |
| ImageRAG | 3.5B | 0.93 | 0.06 | 0.03 | 0.37 | 0.01 | 0.03 | 0.24 |
| **Janus-Pro** | | | | | | | | |
| + RA-CM3 | 1.0B | 0.98 | 0.78 | 0.41 | 0.84 | 0.42 | 0.49 | 0.65 (-0.06) |
| + DAiD (ours) | 1.0B | 0.98 | 0.82 | 0.54 | **0.87** | 0.63 | 0.49 | 0.72 (+0.01) |
| + FAiD (ours) | 1.2B | **1.00** | **0.88** | 0.50 | 0.86 | **0.70** | **0.73** | **0.78** (+0.07) |

Table 1: Evaluation of text-to-image generation ability on GenEval benchmark. Note our methods are based on Janus-Pro highlighted in gray.

for evaluating spatial and frequency coherence through wavelet packet coefficients. For all three metrics, lower scores indicate higher quality generated images. Detailed descriptions of these benchmarks and metrics can be found in Appendix B.4.

# 5 Results and Discussion

## 5.1 Text-to-Image Generation Results

Tables 1, 2, and 3 present performance comparisons across multiple benchmarks, where our AR-RAG methods consistently outperform existing approaches. Notably, previous retrieval-augmented approaches such as RDM and ImageRAG perform worse than their non-retrieval counterparts (LDM and SDXL, respectively) on both GenEval and DPG-Bench. We provide detailed analysis for existing image-level retrieval methods and highlight the unique advantages of our AR-RAG frameworks in the following discussion and Section 5.2. Appendix C.1 provides a benchmark analysis to demonstrate the effectiveness of patch-level retrieval in our AR-RAG methods.

On GenEval, our methods show significant improvements in categories such as "Two Obj." and "Position," which demand accurate multi-object generation and spatial arrangement. These gains are largely due to the local and dynamic nature of our autoregressive patch-level retrieval. Consider the prompt "*a green couch and an orange umbrella*", a combination that rarely co-occurs in real-world images. Static full-image retrieval methods may retrieve references containing only one of the objects.

| Method | Params | Global | Entity | Attribute | Relation | Other | Overall ↑ |
|---|---|---|---|---|---|---|---|
| *Non Retrieval-Augmented Model* | | | | | | | |
| PixArt-$\alpha$ | 0.6B | 74.97 | **97.32** | 78.60 | 82.57 | 76.96 | 71.11 |
| SDv1.5 | 0.9B | 74.63 | 74.23 | 75.39 | 73.49 | 67.81 | 63.18 |
| Janus-Pro | 1.0B | 81.76 | 84.53 | 84.34 | 92.22 | 75.20 | 77.26 |
| Lumina-Next | 2.0B | 82.82 | 88.65 | **86.44** | 80.53 | **81.82** | 74.63 |
| SDXL | 3.5B | 83.27 | 82.43 | 80.91 | 86.76 | 80.41 | 74.65 |
| *Retrieval-Augmented Model* | | | | | | | |
| RDM | 1.4B | 62.36 | 40.46 | 60.20 | 69.16 | 24.68 | 26.51 |
| ImageRAG | 3.5B | 61.35 | 32.77 | 53.87 | 60.38 | 18.42 | 19.82 |
| **Janus-Pro** | | | | | | | |
| + RA-CM3 | 1B | 81.76 | 81.03 | 83.32 | 90.60 | 70.80 | 73.76 (-3.50) |
| +DAiD (ours) | 1.0B | **83.58** | 84.46 | 84.76 | 91.49 | 76.40 | 77.88 (+0.62) |
| +FAiD (ours) | 1.2B | 82.67 | 85.80 | 85.38 | **92.30** | 76.80 | **79.36** (+2.10) |

Table 2: Evaluation of text-to-image generation ability on DPG-Bench. Note our methods are based on Janus-Pro highlighted in gray.

Taking these references as a global visual prior throughout the generation can lead the model to overfit to irrelevant layouts or dominant visual structures in the retrieved examples. On DPG-Bench, which features dense and highly detailed prompts, the performance gap between our method and prior retrieval-augmented approaches becomes even more substantial. Similar as GenEval, existing image-level retrieval augmentation methods struggle to retrieve meaningful references when the number of distinct entities and attributes in a prompt increases. In contrast, our autoregressive augmentation framework overcomes this limitation by dynamically retrieving patch-level visual features based on the evolving image context rather than the original prompt, enabling more targeted and effective augmentation.

On Midjourney-30K, our proposed methods consistently outperform both Janus-Pro and Show-o baselines across all three evaluation metrics. Notably, despite operating locally at the patch level, our approach leads to a significant reduction in FID scores, indicating improved global visual quality and closer alignment with the distribution of real images. This suggests that context-aware, autoregressive retrieval and refinement can propagate to enhance holistic image fidelity. Furthermore, the improvements in CMMD and FWD metrics confirm our method's effectiveness in reducing visual distortions and enhancing coherence. These results also demonstrate that AR-RAG delivers robust and architecture-agnostic improvements, validating its broad applicability across different image generation backbones.

| Model | Params | CMMD ↓ | FID ↓ | FWD ↓ |
|---|---|---|---|---|
| RDM | 1.4B | 0.71 | 19.17 | 34.95 |
| ImageRAG | 3.5B | 0.32 | 19.39 | 62.65 |
| **Show-o** | 1.3B | 0.09 | 11.47 | 2.57 |
| + DAiD (ours) | 1.3B | 0.08 | 9.28 | 2.49 |
| + FAiD (ours) | 1.5B | **0.06** | 7.93 | **1.73** |
| **Janus-Pro** | 1.0B | 0.12 | 14.33 | 28.41 |
| + RA-CM3 | 1.0B | 0.13 | 12.40 | 20.57 |
| + DAiD (ours) | 1.0B | 0.11 | 9.15 | 28.00 |
| + FAiD (ours) | 1.2B | 0.07 | **6.67** | 9.40 |

Table 3: Evaluation of text-to-image generation ability on the Midjourney-30K benchmark.

Tables 4 and 5 present our results on RareBench [29] and T2I-Bench [19]. On RareBench, our AR-RAG methods demonstrate substantial improvements over baselines, particularly in multi-object composition categories such as "Concat," "Relation," and "Complex," where FAiD achieves 67%, 54%, and 65.5% respectively, significantly outperforming previous retrieval-augmented methods. These gains highlight our method's effectiveness in generating rare or novel combinations by dynamically retrieving relevant patch-level features rather than relying solely on model memorization. On T2I-Bench, our methods outperform the Janus baseline across most categories, with particularly notable improvements in "Color" (0.703 vs. 0.652) and "Numeracy" (0.537 vs. 0.251). While the diffusion-based RDM achieves competitive performance on certain sub-categories such as "Texture," this reflects the inherent advantage of continuous diffusion models over discrete token-based autoregressive approaches, as the quantization process inevitably incurs information loss when converting between discrete image tokens and continuous image features. Nonetheless, our autoregressive approach demonstrates strong capability in attribute binding and counting tasks, validating the effectiveness of patch-level retrieval augmentation within the discrete token generation paradigm.

| Method | Params | Color | Shape | Texture | Numeracy |
|---|---|---|---|---|---|
| RDM | 1.4B | 0.666 | **0.555** | **0.538** | 0.382 |
| ImageRAG | 3.5B | 0.670 | 0.543 | 0.528 | 0.344 |
| **Janus-Pro** | 1.0B | 0.652 | 0.455 | 0.449 | 0.251 |
| + DAiD (ours) | 1.0B | 0.630 | 0.459 | 0.464 | 0.245 |
| + FAiD (ours) | 1.2B | **0.703** | 0.494 | 0.483 | **0.537** |

Table 4: Evaluation of text-to-image compositional reasoning on T2I-Bench.

| Method | Params | Property | Shape | Texture | Action | Complex (single obj) | Concat (multi obj) | Relation (multi obj) | Complex (multi obj) |
|---|---|---|---|---|---|---|---|---|---|
| RDM | 1.4B | 68.0 | 56.9 | 56.0 | 44.0 | 64.5 | 56.2 | 49.5 | 39.5 |
| SDXL | 3.5B | 60.0 | 56.9 | 71.3 | 47.5 | 58.1 | 39.4 | 35.0 | 47.5 |
| ImageRAG | 3.5B | 25.0 | **86.5** | **85.5** | 49.0 | 77.0 | 57.5 | 36.0 | 55.5 |
| **Janus-Pro** | 1.0B | 53.5 | 59.0 | 76.5 | 53.5 | 78.0 | 42.5 | 48.0 | 50.5 |
| + DAiD (ours) | 1.0B | 64.0 | 61.5 | 74.0 | 54.0 | 78.5 | 46.5 | 50.5 | 54.5 |
| + FAiD (ours) | 1.2B | **74.0** | 62.0 | 78.5 | **58.5** | **79.0** | **67.0** | **54.0** | **65.5** |

Table 5: Evaluation of text-to-image compositional reasoning on RareBench.

## 5.2 Qualitative Analysis

Figure 4 illustrates these quantitative improvements with representative examples from DPG-Bench (left three columns) and GenEval (right two columns). These examples demonstrate how autoregressive retrieval augmentation improves the vanilla image generation models. The vanilla model struggles with *object interactions* (e.g., column 3, where shoes merge with a coffee machine in the background), *complex structures* (e.g., columns 2 and 5, where camels and sheep have anatomically incorrect numbers of organs), and *implausible configu-*

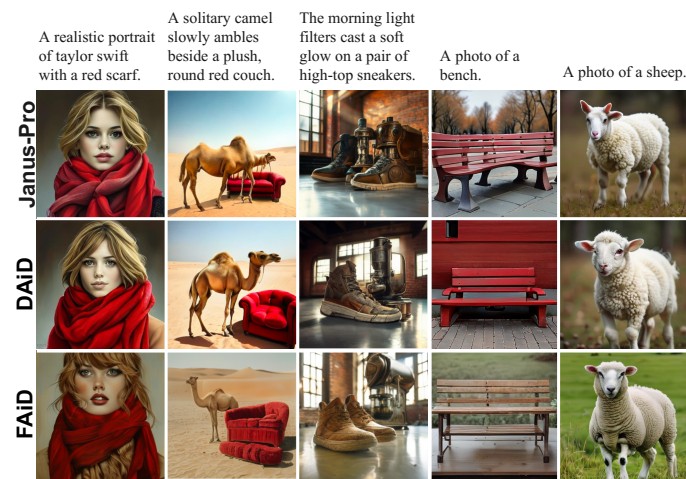

Figure 4: Qualitative results of DAiD, FAiD and baselines.

*rations* (e.g., column 4, where a chair exhibits an impossible design). Both DAiD and FAiD substantially reduce such local distortions, with FAiD yielding the highest visual quality. These results confirm that autoregressive retrieval effectively maintains object consistency and structural integrity throughout the generation process, particularly for complex objects and multi-object scenes.

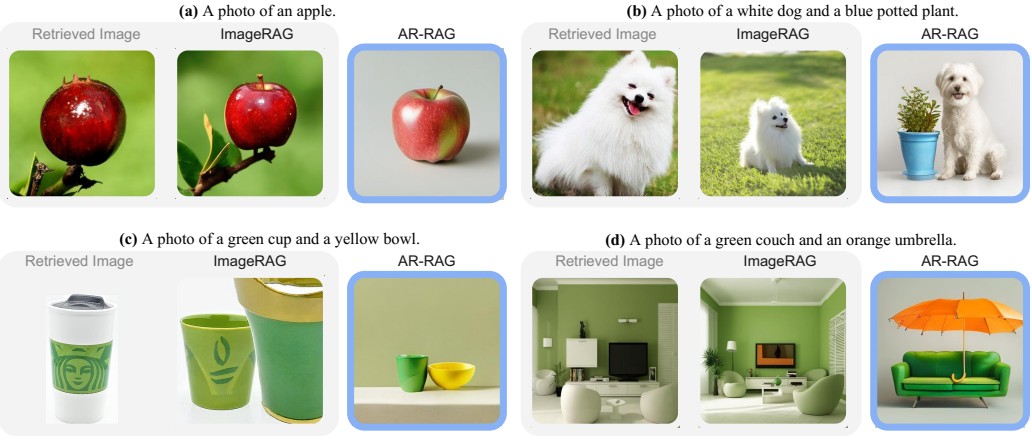

Figure 5: Images generated by ImageRAG [35] and our AR-RAG. ImageRAG excessively copies retrieved images and does not follow user prompts.

Figure 5 presents a comparative analysis of conventional image-level and our autoregressive patch-level retrieval augmentation methods. By comprehensively examining images produced by ImageRAG alongside their corresponding retrieved reference images, we identify two critical challenges inherent in image-level retrieval augmentation approaches. First, these methods tend to overcopy irrelevant visual elements from retrieved reference images into the generation outputs. As illustrated in Figure 5 (a), when generating an image of an apple, image-level retrieval approaches retrieve a reference image showing an apple on a tree branch and subsequently incorporate both the apple and the surrounding branches, despite the prompt making no mention of them. Similarly, for the prompt "*a green cup and a yellow bowl*" in Figure 5 (b), the image-level retrieval augmentation approach retrieves a green Starbucks cup and reproduces the pattern on the cup in the generated image, despite this element not being part of the original instruction. This over-copying behavior directly compromises the instruction-following capability of generative models. Figure 5 (c) demonstrates that when prompted to generate "*A photo of a white dog and a blue potted plant,*" image-level retrieval

| Model | Single GPU (L40) | |
| | Total (s) | Average (s) |
|---|---|---|
| ImageRAG | 879.64 | 8.80 |
| Janus-Pro | 457.74 | 4.58 |
| + DAiD | 459.34 | 4.59 (+0.22%) |
| + FAiD | 623.01 | 6.23 (+36.03%) |

Table 6: Inference time for generating 100 images on a single L40 card.

methods produce an image containing only the white dog, omitting the blue potted plant entirely. Similarly, for "*a photo of a green couch and an orange umbrella*" in Figure 5 (d), the generated image fails to include the umbrella. This degradation in instruction following occurs because image-level retrieval biases the generation process toward the compositional structure of retrieved reference images, which may not align with the multi-object relationships specified in the prompt. In contrast, by autoregressively retrieving and integrating visual information at the fine-grained patch level rather than the image level, AR-RAG enables selective incorporation of relevant visual elements while maintaining independence from irrelevant contextual features present in the reference images.

### 5.3 Inference Time Cost

Table 6 shows the inference time comparisons across different models when generating 100 images using both a single L40 GPU. The DAiD method introduces only a minimal increase in inference time compared to the base Janus-Pro-1B model, with an average overhead of just 0.22%, demonstrating that DAiD maintains high computational efficiency. FAiD shows a more noticeable overhead of 36.03% on a single GPU due to its autoregressive retrieval and feature blending operations. However, this increase remains reasonable given the substantial performance gains in generation quality. Overall, both DAiD and FAiD do not significantly compromise the inference efficiency of Janus-Pro, making them practical for real-world applications.

## 6   Related Work

Retrieval-augmented generation (RAG) has emerged as a powerful paradigm that enhances generative models by incorporating external knowledge during decoding [24, 13, 16, 49, 48, 15, 25, 26, 44]. Originally developed for natural language processing, RAG enables models to retrieve relevant documents to supplement parametric knowledge during response generation [4], and has been widely adopted in many downstream tasks, such as knowledge-intensive tasks [24], document fusion [20], model pretraining [16], dialogue generation [37, 1], and so on.

Beyond the text domain, prior research has explored enhancing image generation by incorporating external visual references. Early approaches [8, 3] condition the diffusion process on retrieved images, typically encoded via CLIP or VAE encoders, to guide generation toward higher visual fidelity. KNN-Diffusion [36] extends this idea by leveraging $k$-nearest neighbor images to improve zero-shot generalization to novel domains. Building on this retrieval-augmented framework, more recent methods [51, 35] introduce adaptive retrieval pipelines that iteratively refine retrieved images based on feedback from multimodal large language models (MLLMs) analyzing the generated outputs. These methods enable context-aware and prompt-sensitive guidance during generation. Another line of work [48] encodes multimodal retrievals into discrete visual and text tokens, and uses them directly as contextual input to augment the generation process of a multimodal large language model. All of these works differ from our method by that our method works on patch-level, enabling more fine grain retrievals and can dynamically adjust retrievals based on evolving generation states.

## 7   Conclusion

In this work, we propose Autoregressive Retrieval Augmentation (AR-RAG), a novel retrieval paradigm that enhances image synthesis by leveraging k-nearest neighbor retrievals at the patch level. Unlike traditional image-level retrieval approaches, AR-RAG enables fine-grained visual element integration while maintaining compositional flexibility. We introduce two parallel frameworks: (1) Distribution-Augmentation in Decoding (DAiD), a training-free approach that integrates retrieved patch distributions directly into generation, and (2) Feature-Augmentation in Decoding (FAiD), which employs parameter-efficient fine-tuning with multi-scale feature smoothing and compatibility-based feature augmentation. Extensive experiments across GenEval, DPG-Bench, and Midjourney-30K demonstrate that AR-RAG significantly outperforms both conventional and retrieval-augmented baselines, particularly in handling complex prompts with multiple objects and specific spatial relationships. Our methods substantially reduce local distortions in generated images, improving object consistency and structural integrity.

### Acknowledgment

This research is supported by the award No. #2238940 from the Faculty Early Career Development Program (CAREER) of the National Science Foundation (NSF). The views and conclusions contained

herein are those of the authors and should not be interpreted as necessarily representing the official policies, either expressed or implied, of the U.S. Government. The U.S. Government is authorized to reproduce and distribute reprints for governmental purposes notwithstanding any copyright annotation therein.

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

# A   Multi-Scale Feature Smoothing Algorithm

Algorithm A illustrates the multi-scale feature smoothing, which is the core computational procedure for refining retrieved patch representations within their generation context. This algorithm ensures that retrieved visual elements are spatially and stylistically coherent with the surrounding image content through systematic multi-scale convolution operations.

The algorithm processes each retrieved patch representation $\hat{\mathbf{h}}_i$ independently, applying convolution operations at multiple scales ranging from $2 \times 2$ to $Q \times Q$ kernels. For each scale $q$, the algorithm initializes a temporary feature tensor $\mathbf{M} \in \mathbb{R}^{Q \times Q \times D}$ and an accumulation vector $\hat{\mathbf{h}}_q \in \mathbb{R}^D$. The nested loops over indices $m$ and $n$ systematically extract local patch features from different spatial windows around the target position $(i, j)$. Each extraction operation $\mathbf{H}^l_{\text{loc}} \leftarrow \mathbf{H}^l[i - m : i + q - m, j - n : j + q - n]$ captures a local neighborhood of size $q \times q$ centered at varying offsets from the target position.

---

**Algorithm 1:** Multi-Scale Feature Smoothing

**Input:** Image Representations $\mathbf{H}^l \in \mathbb{R}^{\sqrt{N} \times \sqrt{N} \times D}$,
   Retrieved Patch Representations
   $[\hat{\mathbf{h}}_1, \hat{\mathbf{h}}_2, \ldots, \hat{\mathbf{h}}_K]$, Next Patch Index $(i, j)$
**Output:** Updated hidden states $[\hat{\mathbf{h}}_1, \hat{\mathbf{h}}_2, \ldots, \hat{\mathbf{h}}_K]$

1  **foreach** $\hat{\mathbf{h}}_i \in [\hat{\mathbf{h}}_1, \ldots, \hat{\mathbf{h}}_K]$ **do**
2      **for** $q = 2$ ***to*** $Q$ **do**
3          Initialize tensor: $\mathbf{M} \leftarrow \mathbf{0} \in \mathbb{R}^{Q \times Q \times D}$;
4          Initialize tensor: $\hat{\mathbf{h}}_q \leftarrow \mathbf{0} \in \mathbb{R}^D$;
5          **for** $m = q$ ***down to*** $1$ **do**
6              **for** $n = q$ ***down to*** $1$ **do**
7                  $\mathbf{H}^l_{\text{loc}} \leftarrow \mathbf{H}^l[i - m : i + q - m, j - n : j + q - n]$;
                   $\mathbf{M}_{mn} \leftarrow \text{Conv}^1_{q \times q}(\mathbf{H}^l_{\text{loc}})$;
9          $\hat{\mathbf{h}}_q \mathrel{+}= \text{Conv}^2_{q \times q}(\mathbf{M})$;
10     $\hat{\mathbf{h}}_i \leftarrow \frac{\hat{\mathbf{h}}_q}{Q-1}$;

---

The extracted local features undergo two-stage convolution processing. The first convolution operation $\text{Conv}^1_{q \times q}$ transforms the local patch features into an intermediate representation stored in $\mathbf{M}_{mn}$, effectively capturing contextual relationships within each local window. The second convolution operation $\text{Conv}^2_{q \times q}$ processes the accumulated intermediate features to produce scale-specific refined representations. This two-stage design enables the algorithm to first capture local contextual patterns and then integrate them into a coherent scale-specific feature representation.

After processing all scales for a given retrieved patch, the algorithm computes the final refined representation by averaging the scale-specific features. The normalization factor $(Q - 1)$ accounts for the number of scales processed, ensuring consistent feature magnitudes across different retrieved patches. This averaging operation effectively combines multi-scale contextual information into a single refined representation that preserves both fine-grained details from smaller kernel sizes and broader contextual patterns from larger kernel sizes. The resulting refined patch representations maintain spatial coherence with the surrounding generation context while preserving the essential visual characteristics of the retrieved content.

# B   Experiment Setup

## B.1   RA-CM3 Implementation Details

Since the pretrained RA-CM3 model is not publicly available, we implement our own version following the methodology described in the original paper to serve as a representative baseline for image-level retrieval-augmented generation. Our implementation uses Janus-Pro as the backbone model to ensure fair comparison with our proposed methods, as both approaches operate on the same foundation architecture.

We construct an image-level retrieval database using the same CC12M [5] and JourneyDB [38] datasets employed for our patch-level retrieval database to maintain consistency in the underlying data distribution. All images in the database are encoded into 512 dimensional vector representations using a pretrained CLIP [31] model. For each training instance in our 50,000 sample training set, we retrieve the most relevant reference image by encoding the corresponding text prompt with the same CLIP model, extracting the [**CLS**] token as the text representation, and computing cosine similarity scores between the text representation and all image representations in the database. The image

with the highest similarity score is selected as the retrieved reference. Each retrieved image is then processed through the quantized autoencoder from Janus-Pro to obtain image tokens $[v_1, \ldots, v_N] = \mathcal{Z}(\theta_{Enc}(I))$, which are subsequently encoded into 2048 dimensional vector representations in the language model's latent space using the image embedding and aligning layers in Janus-Pro. These retrieved image representations are concatenated with the text embeddings of the input prompts to form the augmented input for training the retrieval-enhanced model, which is the same training strategy used in RA-CM3.

During inference, given a text prompt for image generation, we follow the same retrieval process used in training. The input prompt is encoded using the CLIP text encoder, and we compute cosine similarity with all images in the database to identify the most relevant reference image. The retrieved image is processed through the same pipeline to obtain its representation in the language model's latent space. This representation is then prepended to the text prompt embedding to provide the model with both textual and visual context for generation. The augmented input is fed into the fine-tuned Janus-Pro model to generate the output image following the standard autoregressive generation procedure.

### B.2 Show-o Implementation Details

Our patch-based autoregressive retrieval augmentation methods can be theoretically adapted to any model that generates images through discrete tokens. To demonstrate this generalizability, we implement both DAiD and FAiD on the Show-o [46] model, which generates images through a masked token decoding process rather than strict left-to-right autoregression. Show-o decodes multiple image tokens simultaneously at each time step by converting masked tokens to specific image tokens based on a learned probability matrix. This fundamental difference in generation strategy necessitates several architectural adaptations to effectively incorporate our patch-based retrieval mechanisms while maintaining the model's inherent generation capabilities.

**DAiD on Show-o** The implementation of DAiD on Show-o requires three key modifications to accommodate its non-autoregressive generation strategy. First, instead of constructing retrieval queries from upper-left neighboring patches as in autoregressive models, we utilize all eight surrounding patches to form the $h$-hop neighborhood representation for each target token position $(i, j)$. This comprehensive neighborhood encoding is computed as $[\mathbf{V}_{(i-1)(j-1)} : \mathbf{V}_{(i-1)(j)} : \mathbf{V}_{(i-1)(j+1)} : \mathbf{V}_{(i)(j-1)} : \mathbf{V}_{(i)(j+1)} : \mathbf{V}_{(i+1)(j-1)} : \mathbf{V}_{(i+1)(j)} : \mathbf{V}_{(i+1)(j+1)}]$, where missing positions are filled with zero vectors $\mathbf{0}$. Second, to mitigate retrieval noise arising from sparse neighborhood information in early time steps, we apply patch-level retrieval only during the final half of Show-o's decoding process when sufficient contextual information is available. Third, since Show-o simultaneously predicts tokens for all patch positions at each time step rather than sequentially, we perform retrieval for all patch positions concurrently. At each qualifying time step $t$, for every patch position $(i, j)$ in the partially generated image, we extract the eight-neighborhood representation as the retrieval query and obtain the top-$K$ most similar patches $[\hat{\mathbf{v}}_1^{(i,j)}, \hat{\mathbf{v}}_2^{(i,j)}, ..., \hat{\mathbf{v}}_K^{(i,j)}]$ from our database. We then construct position-specific retrieval distributions $D_{\text{retrieval}}^{(i,j)} \in \mathbb{R}^{|\mathcal{Z}|}$ using the same softmax formulation over retrieval distances as described in the main paper. These retrieval distributions are merged with Show-o's predicted distributions for each patch position using the weighted average $D_{\text{merge}}^{(i,j)} = (1 - \lambda) \cdot D_{\text{model}}^{(i,j)} + \lambda \cdot D_{\text{retrieval}}^{(i,j)}$, where $\lambda$ controls the retrieval influence across all positions.

**FAiD on Show-o** The adaptation of FAiD to Show-o involves both training and inference modifications to accommodate the model's masked token generation process. During training, we prepare the training dataset by applying Show-o's noise injection process to generate intermediate noisy representations at each time step, which serve as ground truth targets for the denoising process. For each training instance, we save these intermediate representations and apply patch-level retrieval to obtain relevant patches for all time steps. The training objective remains consistent with the standard Show-o formulation, but with augmented input representations that incorporate retrieved patch information. We insert FAiD modules into every $L/b$ decoder layers of Show-o's $\Phi$ model, where each module processes all patch positions simultaneously rather than focusing on a single next token. At each qualifying time step and for each FAiD-equipped layer $l$, we construct the 2D spatial representation $\mathbf{H}^l \in \mathbb{R}^{\sqrt{N} \times \sqrt{N} \times D}$ from the current hidden states and perform multi-scale feature smoothing for all patch positions. For each position $(i, j)$ and its corresponding retrieved patches

$[\hat{\mathbf{h}}_1^{(i,j)}, \hat{\mathbf{h}}_2^{(i,j)}, ..., \hat{\mathbf{h}}_K^{(i,j)}]$, we apply the convolution operations $\{\text{Conv}_{2\times2}, \text{Conv}_{3\times3}, ..., \text{Conv}_{Q\times Q}\}$ to capture contextual patterns at multiple scales. The refined representations are computed as $\hat{\mathbf{h}}_k^{(i,j)} \leftarrow \sum_{q=2}^{Q} \text{softmax}(\boldsymbol{\Omega})_q \cdot \hat{\mathbf{h}}_{k,q}^{(i,j)}$, where $\hat{\mathbf{h}}_{k,q}^{(i,j)}$ represents the output of the $q \times q$ convolution for patch $k$ at position $(i,j)$. The final augmented representation for each position is calculated as $h_{ij}^{(l+1)} = h_{ij}^l + \Delta h_{ij}^l + \sum_{k=1}^{K} s_k^{(i,j)} \hat{\mathbf{h}}_k^{(i,j)}$, where $\Delta h_{ij}^l$ represents the standard transformer layer updates including self-attention and feed-forward components, and $s_k^{(i,j)}$ are position-specific compatibility scores computed through learned linear projections. During inference, we follow the same procedure but apply retrieval and feature blending only during the final half of the generation time steps to ensure sufficient contextual information is available for effective patch integration.

## B.3 Training Setup

**Training Datasets** For model training, we utilize two large-scale image-caption datasets: CC12M [5] and Midjourney-v6 [5]. From the training sets of these datasets, we randomly sample a total of $50,000$ image-caption pairs ($25,000$ from each dataset) to fine-tune our model. Each image is encoded into $576$ patch features and corresponding image tokens with the same image tokenizer [39] employed in the Janus-Pro model. For each image patch, we further retrieve the top-$K$ image tokens from our retrieval database that exhibit similar neighborhood relationships. Consequently, each training instance comprises: (1) a textual image caption that serves as the conditioning input, (2) a sequence of $576$ image tokens representing the ground-truth image, where each token is paired with $K$ relevant image tokens retrieved from the database based on similar contextual features.

**Training Details** For the implementation of our FAiD approach, we fine-tune two pre-trained text-to-image generation models using the training dataset of 50K text-image pairs that we constructed. We select Janus-Pro-1B [9] and Show-o [46] as our base models. The fine-tuning process is conducted on 4 NVIDIA A100 (80GB) GPUs with a global batch size of 256 for a single epoch. We utilize the AdamW optimizer without weight decay, incorporating a 10% linear warm-up schedule followed by a constant learning rate of 2e-4.

## B.4 Evaluation Benchmarks and Metrics

To comprehensively evaluate our proposed methods, we adopt multiple widely used benchmarks that assess different aspects of image generation quality:

- **GenEval** [14] is a benchmark designed to evaluate models' ability to understand and generate images based on specific attributes and relationships described in text prompts. It comprises multiple categories such as single object generation, two-object composition, counting, colors, positioning, color attribution and so on. Performance is measured as the percentage of generated images that correctly align with the text descriptions.

- **DPG-Bench** [18] (Detailed Prompt Generation Benchmark) evaluates how well image generation models handle detailed prompts with complex requirements, covering categories such as global image quality, entity generation, attribute accuracy, relationship modeling, and other complex generation tasks. Scores are reported as percentages.

- For the **Midjourney-30k benchmark** [42], we employ three complementary metrics to evaluate the quality of generated images, including (1) Fréchet Inception Distance (FID) [17], which measures the statistical similarity between the distributions of generated and real images in the feature space of a pre-trained Inception network; (2) CLIP-MMD (CMMD) [21], which measures the distance between real and generated images using CLIP embeddings and the Maximum Mean Discrepancy, and is specifically designed to better align with human perception of image quality and addresses several limitations of FID, including poor sample efficiency and incorrect normality assumptions; and (3) Fréchet Wavelet Distance (FWD) [41], which measures the distance between real and generated images in the wavelet packet coefficient space. FWD captures both spatial and frequency information without relying on pre-trained networks, making it domain-agnostic and robust to domain shifts across various image types. *For all three metrics, lower scores indicate higher-quality image generation, with both CMMD and FWD particularly effective in capturing distortions in generated images in ways that better correlate with human judgements.*

---

[5] https://huggingface.co/datasets/brivangl/midjourney-v6-llava

## C   Experiment Results and Discussion

### C.1   Accuracy of Patch-based Autoregressive Retrieval

To assess the effectiveness of our patch-level autoregressive retrieval mechanism, we conduct a comparative analysis between the top-$K$ retrieved image tokens and the ground-truth tokens to be generated. Specifically, we randomly sampled $1,000$ instances from our training set, each comprising $576$ image tokens and $576 \times k$ retrieved tokens. To demonstrate the accuracy of the retrieved image tokens, for each ground-truth image token, we also randomly sample a vocabulary code as non-relevant tokens. Using the shared codebook, we transform all image tokens into vector representations and compute the $l_2$ distances between each ground-truth image token and its top-$K$ retrieved counterparts. Similarly, we also compute the mean of the $l_2$ distance between each ground-truth token and the randomly sampled tokens. As shown in Figure 6, the $l_2$ distance between retrieved tokens and ground-truth image tokens is significantly smaller than the distance between randomly sampled tokens and ground-truth tokens. As $k$ increases, the distance between the $k$-th retrieved token and the ground-truth token also increases, demonstrating the effectiveness of the retrieval approach and our assumption that image patches with similar neighbors usually exhibit inherent similarities.

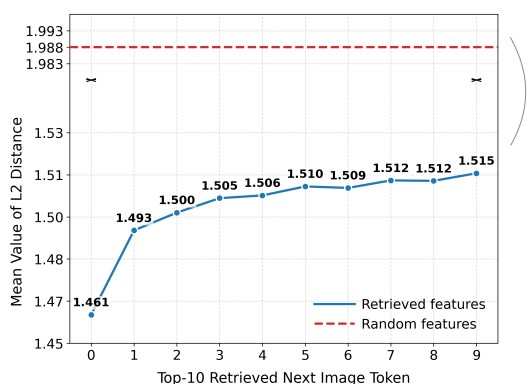

Figure 6: $l_2$ distance between ground-truth tokens and top-10 retrieved tokens (blue line) compared to randomly sampled tokens (red dashed line). The curved arrow indicates a broken y-axis that accommodates the large gap between the retrieved token and the random token baseline.

### C.2   Hyperparameter Optimization

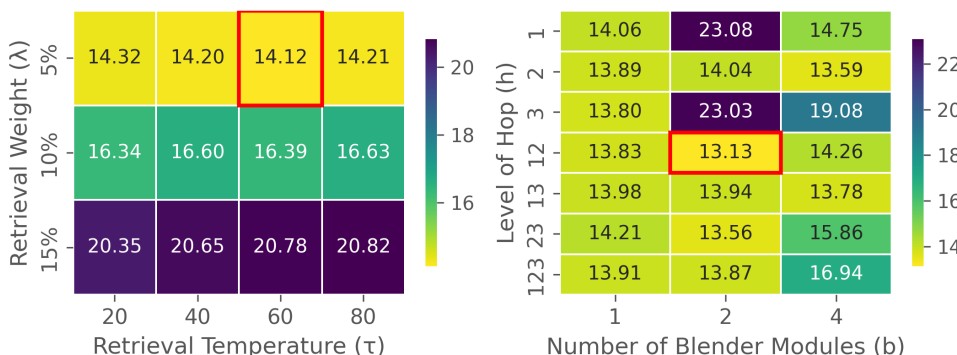

Figure 7: Hyperparameter optimization results for DAiD and FAiD on FID scores. Left: FID scores for DAiD across different combinations of retrieval temperature $\tau$ and merging weight $\lambda$. Right: FID scores for FAiD across varying levels of hop $h$ and numbers of blender modules $b$. All experiments conducted on the Midjourney-10K benchmark, with optimal configurations highlighted by red borders.

Both DAiD and FAiD require careful optimization of distinct sets of hyperparameters. For DAiD, we optimize the retrieval temperature $\tau$ and merging weight $\lambda$, which control the retrieval-based probability distribution sharpness and the balance between retrieval and model predictions, respectively. For FAiD, we focus on the level of hop ($h$) and number of blender modules ($b$), determining the spatial context incorporated during retrieval and extent of feature blending. To identify optimal

configurations, we conducted a systematic ablation study on the Midjourney-10K benchmark using Fréchet Inception Distance (FID) as the performance metric.

Figure 7 presents the FID scores for DAiD across different combinations of $\lambda$ and $\tau$, and for FAiD across varying levels of ($h$) and ($b$), where composite hop levels such as "12" indicate combined use of multiple hop distances. Analysis of the DAiD results reveals that performance degrades as $\lambda$ increases, suggesting that modest integration of retrieval information enhances performance while excessive reliance impairs generative flexibility. The retrieval temperature $\tau$ demonstrates less pronounced effects, though a moderate value of $0.6$ provides marginal benefits. For FAiD, configurations incorporating multiple hop levels generally outperform single hop levels, with the "12" configuration yielding optimal results. Regarding blender modules, an intermediate value consistently delivers the best performance, implying that moderate feature blending optimizes the incorporation of retrieved patches while avoiding both under-utilization and over-smoothing. Based on this analysis, we selected $\lambda = 0.05$) and $\tau = 0.6$ for DAiD, and hop levels "12" with 2 blender modules for FAiD, achieving FID scores of $14.12$ and $13.13$, respectively. These configurations effectively harness retrieval information while preserving the generative strengths of the underlying Janus-Pro model, as demonstrated by their superior performance on the benchmark.

## D  Limitations

While our AR-RAG framework demonstrates strong performance across multiple benchmarks, several limitations should be acknowledged. First, our approach relies on discrete image tokenization and targets discrete token-based models, so it may not be directly applied to continuous diffusion models operating in latent spaces. Second, due to computational resource limitations, our retrieval database remains smaller than billion-scale databases. This limitation may introduce visual pattern biases, as the database may not fully capture the diversity of real-world visual patterns, potentially affecting the generation of underrepresented visual elements. Third, our implementation focuses exclusively on 2D image generation. While the underlying patch-based retrieval concept could theoretically extend to other structured generation tasks such as 3D point cloud generation, we have not explored these applications.

## E  Broader impacts

We propose a novel retrieval-augmented approach to enhance existing image generation models. Our method is both highly efficient and readily adaptable to a wide range of applications, making it valuable for both academic research and industrial deployment. However, as our approach builds upon existing generative models, it may inherit their biases and could potentially produce inappropriate outputs in the absence of additional safety mechanisms. Furthermore, the large-scale retrieval database may contain unsafe or undesirable content, which can be reflected in the retrieved image patches. To ensure safe deployment in real-world scenarios, additional safeguards and filtering measures are necessary to mitigate these risks.

