# OpenReview forum: "AR-RAG: Autoregressive Retrieval Augmentation for Image Generation"
_NeurIPS.cc/2025/Conference — NeurIPS 2025 poster_

### Official Review · Reviewer_L36Z · 2025-06-29

**Clarity:** 3
**Significance:** 3
**Originality:** 3
**Rating:** 5
**Confidence:** 5

**Summary:**

This paper proposes a method that can retrieve image patches for image generation. Using the fact that the auto-regressive generation model (Janus-pro) utilizes a quantized image tokenizer, the algorithm finds the top-k relevant patches and modify the distribution from which each location is sampled. DaiD directly uses the image patches and FaiD softly integrates the retrieved patches to the model.

**Questions:**

No further questions.

**Ethical Concerns:**

["NO or VERY MINOR ethics concerns only"]

**Final Justification:**

The authors resolve my comment about zero-shot generation and I am pleased that the result came out as I expected. I believe this work might be a quite important contribution for image generation models since it can retrieve per-image contribution, which might give a clue to solve the copyright issue for data providers. Therefore, I maintain my score as is.

**Limitations:**

yes

**Quality:**

3

**Strengths And Weaknesses:**

Strengths
- This method is obviously novel and technically solid. I have seen many works that give reference images to the model but exploitation on image patches seems to be something new. This reminds me of PatchMatch algorithm which works really well on images with complex texture in Photoshop.
- Injecting inductive bias to image generation models is usually very unstable and hard to handle, but this method is handling this problem with a clever method.

Weaknesses
- I really want to see if this method works on a problem setting such as zero-shot image generation. For example, generating a specific object or person often needs few images and LoRA does the rest. Auto-regressive models like Janus-pro are capable of doing the same task by accepting reference image but the performance is usually limited if the reference has been unseen during the training. I wonder if this method can cope with this setting and I am very sure this will add significant contribution to this paper.

---

> ### Author Rebuttal · Authors · 2025-07-31
>
> We sincerely thank the reviewer for this insightful suggestion regarding zero-shot image generation and the importance of evaluating performance in such scenarios. In response, we conducted a targeted experiment to assess our method’s capabilities in this setting.
>
> Specifically, we constructed an artificial retrieval database containing 30 images of Taylor Swift, collected from the internet, while ensuring no other human faces were included. The artificial retrieval database also contains 10,000 images from CC12M. (We filtered out images that contain human faces based on their captions) We then used the prompt “a woman is playing guitar on the stage” and generated 10 images each with both DAiD and FAiD. We observed that, when Taylor Swift’s images were present in the retrieval database, the generated faces in most images closely resembled her appearance. Conversely, after removing Taylor Swift’s images from the database and repeating the generation, the generated faces no longer displayed her features.
>
> These results demonstrate that our retrieval-augmented approach can effectively leverage reference images for specific object or person generation, and is sensitive to the presence of such references in the retrieval set. We agree that further systematic exploration of zero-shot and few-shot settings is a promising direction and will add significant value to future versions of our work. We will make sure to include this new result in the revised version.

---

### Official Review · Reviewer_FaiV · 2025-06-30

**Clarity:** 3
**Significance:** 2
**Originality:** 3
**Rating:** 4
**Confidence:** 5

**Summary:**

This paper proposes a new image generation paradigm, Autoregressive Retrieval Augmentation (AR-RAG), which dynamically integrates fine-grained visual content by autoregressively incorporating k-nearest neighbor retrievals at the patch level. This approach avoids the issue of over-copying full-image information in existing image RAG methods. To implement this framework, the authors introduce DAiD, a training-free plug-and-play decoding strategy, and FAiD, a parameter-efficient fine-tuning method. Experiments on benchmarks such as Midjourney-30K, GenEval, and DPG-Bench validate the effectiveness of AR-RAG.

**Questions:**

please refer to the weakness part

**Ethical Concerns:**

["NO or VERY MINOR ethics concerns only"]

**Final Justification:**

The authors have resolved my concern. I tend to borderline accept this paper

**Limitations:**

yes

**Quality:**

3

**Strengths And Weaknesses:**

Strength:
1, The validation motivation is reasonable. Compared with previous global Image RAG methods, AR-RAG utilizes region-of-interest (ROI) patch information more rationally.
2. The writing is clear and easy to understand, facilitating comprehension of the proposed approach.

Weakness:
1. The inference speed is notably slow, hindering practical application. Unlike Image RAG, which requires only one retrieval, AR-RAG performs O(N²) retrievals, where N is the number of generated patches. This quadratic complexity significantly increases computational overhead.
2. The experiments are conducted against relatively weak baselines. For example, the "cat playing basketball" issue in the teaser might not exist in the latest methods like FLUX, indicating that the comparative evaluation could be more rigorous.

---

> ### Author Rebuttal · Authors · 2025-07-31
>
> **Question:** The inference speed is notably slow, hindering practical application. Unlike Image RAG, which requires only one retrieval, AR-RAG performs O(N²) retrievals, where N is the number of generated patches. This quadratic complexity significantly increases computational overhead.
>
> **Answer:** We would like to clarify that the retrieval complexity of our method is O(N), not O(N²) as stated in the review. Specifically, for N patches, our approach performs N retrievals (i.e., one per image patch) rather than a quadratic number of retrievals. For each patch, we retain only the top 2,048 retrieved candidates. Furthermore, we employ the Faiss library to construct our retrieval database, which is highly optimized for large-scale and hierarchical nearest neighbor search. As demonstrated in Table 4 of our submission, the average inference time for Janus-Pro + DAiD to generate 100 images increases by only 0.22% compared to the Janus-Pro baseline and most of the addtional inference time is attributed to the retrieval process during inference. This negligible increase underscores the efficiency and practical feasibility of our retrieval method. We hope this clarification addresses the concerns regarding computational overhead.
>
> ---
>
> **Question:** The experiments are conducted against relatively weak baselines. For example, the "cat playing basketball" issue in the teaser might not exist in the latest methods like FLUX, indicating that the comparative evaluation could be more rigorous.
>
> **Answer:** As suggested by the reviewer, we compare our method Janus-Pro + FAiD against two SOTA diffusion models including Flux-dev [1] and SD3-medium [2] on GenEval and MJHQ 30K. As reported in the Table below, these models consistently underperform relative to our proposed method, Janus-Pro + FAiD, despite their larger parameter sizes and pre-training on larger datasets.
>
> | Model                    | GenEval | MJHQ 30K |
> |--------------------------|---------|----------|
> | Janus-Pro + FAiD (1.3B)  | 0.78    | 6.67     |
> | Flux-dev [1] (12B)       | 0.67    | 10.15    |
> | SD3-medium [2] (2B)      | 0.62    | 11.92    |
>
>
> Notably, the GenEval benchmark specifically assesses the ability to generate objects with rare attributes and to compose images containing multiple, less commonly co-occurring objects. The comparatively lower performance of Flux-dev and SD3-medium on these benchmarks suggests that these models have limitations in handling complex generation tasks—such as modeling spatial relations, attribute binding, and multi-object composition—even in their latest versions. We believe this comparative analysis is both fair and rigorous, and it demonstrates the unique strengths of our approach.
>
> ---
>
> [1] Black Forest Labs. Flux, 2024.
>
> [2]  Scaling rectified flow transformers for high-resolution image synthesis.

---

> ### Comment · Reviewer_FaiV · 2025-08-07
>
> Thanks for the authors' answers. My concern is resoloved. I tend to weakly accept this paper now.

---

> > ### Author Response · Authors · 2025-08-08
> >
> > Thank you for your follow-up and for reconsidering your rating. We appreciate your confirmation that our responses have resolved your concerns. We will include the additional comparison experiments with SOTA diffusion models in the revised version of our paper.

---

### Official Review · Reviewer_Dnu4 · 2025-07-03

**Clarity:** 4
**Significance:** 2
**Originality:** 4
**Rating:** 4
**Confidence:** 3

**Summary:**

This paper introduces a novel method that performs retrieval augmentation during the autoregressive image generation process. For each patch, they retrieve likely relevant patches from a visual patch database given previously predicted tokens. They then incorporate these patches into the generation process using distribution augmentation, which biases the next-token distribution given the retrieved next patch, and feature augmentation (feature blending). The paper claims significant improvements on GenEval, DPG-Bench, and Midjourney-30k.

**Questions:**

I mostly want to see more compute controlled baselines. Namely, increase the number of parameters on Janus Pro until inference time matches +FAiD and fine-tune on the same dataset mixtures. Another baseline is just fine-tuning Show-o and Janus Pro on the dataset mixtures directly. As it is, it's hard to disentangle this method from the impact of compute and data. Another use case that I believe would strengthen the paper is the "training free distribution alignment" potential. Seems like including JourneyDB in the retrieval set improves Midjourney-30k with DAiD. Given that personalization is a popular RAG use case, I think this could be a useful supplementary experiment. Again, however, it would have to outperform just fine-tuning.

**Ethical Concerns:**

["NO or VERY MINOR ethics concerns only"]

**Final Justification:**

I move my rating towards acceptance because the authors have provided compute-controlled baselines. Further their SFT with specific datasets provides insight into how much of the gain is from the retrieval dataset mixture versus the retrieval process itself.

**Limitations:**

yes

**Quality:**

3

**Strengths And Weaknesses:**

### Strengths

+ Very creative strategy. This is the first method I've seen that incorporates multiple retrieval steps during the generative process for a single image. Brings to mind RAG-based iterative refinement methods.
+ DAiD (distribution augmentation) strategy is simple and intuitive. Interested to see a simple baseline with DAiD by itself (no model, just D_retrieval).
+ The paper is well-written. The method is complex and the paper does a good job of presenting it in easy to understand chunks.
+ One potential advantage is the fact that DAiD is training-free. Not as touched-upon, but it could work for on the fly adaptation with a user provided DB. Results in Table 3 seem to support this use case.


### Weaknesses

The primary weakness is that it is not clear from the stated results that this method is worth the compute and added complexity. (+DAiD) results are within margin of error for GenEval and DPG-Bench. (+FAiD) adds 20% extra parameters and 35% extra compute (on an L40, presumably this would get worse on faster GPUs since retrieval overhead would be a greater fraction of the forward pass). It's unclear if more parameters + a bit more training would yield the entirety of the gains from FAiD given equal inference time.

For Table 3, t2i generation on Midjourney-30k, I noticed that one of the retrieval datasets and fine-tuning datasets is JourneyDB, which is constructed with Midjourney images and is therefore more likely to be in distribution. I'd like to see if just fine-tuning with extra parameters on JourneyDB is enough to improve the Midjourney-30k scores as a baseline.

---

> ### Author Rebuttal · Authors · 2025-07-31
>
> We appreciate the reviewer's thoughtful feedback regarding computational efficiency and the source of performance improvements in our method.
>
> ---
>
> To demonstrate that the performance improvements stem from the retrieval mechanism rather than merely from distributional similarities between training and evaluation databases, we conducted the suggested experiments that fine-tune the baseline model on JourneyDB and our mixture dataset. The results are presented below:
> | models   | CMMD | FID | FWD |
> |----------|---------| ---------| ---------|
> | Janus 1B |    0.12      |     14.33     |     28.41     |
> | Janus 1B + DAiD |     0.11     |    9.15      |    28.00    |
> | Janus 1B + FAiD |      0.07    |    6.67      | 9.40 |
> | Janus 1B + JourneyDB SFT |     0.09     |   11.47      | 18.76 |
> | Janus 1B + Mixture-Data SFT |     0.12     |   11.94      | 16.96 |
>
> As expected, fine-tuning on in-domain datasets (JourneyDB and the Midjourney training set) improves performance, with JourneyDB fine-tuning outperforming the mixture dataset approach. However, our FAiD method significantly outperforms both fine-tuning approaches across all metrics, demonstrating clear benefits from our retrieval mechanism beyond what can be achieved through domain-specific fine-tuning alone.
>
> ---
>
> Regarding computational overhead, we would like to first clarify that: (1) Our retrieval process is optimized using the FAISS library with GPU acceleration, resulting in minimal overhead. As shown in Table 4, the DAiD method adds only 0.22% additional time compared to the vanilla Janus-Pro model. (2) The primary computational cost in FAiD comes from the additional convolutional operations in the decoder. This overhead is expected to decrease proportionally when running on faster GPU devices.
>
> Furthermore, we agree with the reviewer's suggestion to provide more explicit demonstrations of benefits versus costs. As suggested, we are conducting additional experiments with parameter-augmented baseline models. Specifically, we have added adapters to the vanilla Janus-Pro model and are fine-tuning it with the identical training setting. Due to time constraints, these experiments are still in progress. We commit to sharing these results during the discussion period and incorporating them in the next version of our paper.

---

> > ### Author Response · Authors · 2025-08-03
> >
> > **Question:** I mostly want to see more compute-controlled baselines. Namely, increase the number of parameters on Janus Pro until inference time matches +FAiD and fine-tune on the same dataset mixtures.
> >
> > | models   | CMMD | FID | FWD |
> > |----------|---------| ---------| ---------|
> > | Janus 1B |    0.12      |     14.33     |     28.41     |
> > | Janus 1B + FAiD |      0.07    |    6.67      | 9.40 |
> > | Janus 1B + Adapter |  0.11    |  9.61  | 17.72 |
> >
> > Following the reviewer’s suggestion, we added adapters after both the attention and feedforward modules in each transformer layer of Janus-Pro. Each adapter is set to a dimension of 1024, resulting in approximately 2M additional parameters, matching the number of additional parameters in Janus 1B + FAiD. To ensure a fair comparison, all training hyperparameters and datasets were kept identical to those used for Janus 1B + FAiD. The results are reported in the above table. As observed, Janus 1B + FAiD outperforms Janus 1B + Adapter by a significant margin, highlighting the effectiveness of our proposed autoregressive retrieval augmentation.

---

### Official Review · Reviewer_zZU3 · 2025-07-06

**Clarity:** 3
**Significance:** 3
**Originality:** 4
**Rating:** 5
**Confidence:** 4

**Summary:**

This paper tackles the problem of enhancing image generation by introducing Autoregressive Retrieval Augmentation (AR-RAG), which performs patch-level k-nearest neighbor retrievals at each generation step. Unlike previous static retrieval-based methods, AR-RAG dynamically incorporates context-aware visual references to improve generation quality. The authors propose two decoding strategies: DAiD, a training-free distribution-level augmentation, and FAiD, a fine-tuning method leveraging smoothed patch features. Experimental results on multiple benchmarks show that AR-RAG significantly improves object consistency, structural coherence, and performance on complex prompts.

**Questions:**

While the paper is of high quality overall, incorporating additional benchmark evaluations and deeper analyses, as discussed in the weaknesses section, could further improve its completeness and impact.

**Ethical Concerns:**

["NO or VERY MINOR ethics concerns only"]

**Final Justification:**

I have thoroughly checked the authors’ rebuttals as well as the reviews from other reviewers. The responses sufficiently resolve my initial concerns on performance on other benchmarks and analysis on GPU memory. I appreciate the authors’ efforts and will keep my original recommendation of “accept.”

**Limitations:**

yes

**Paper Formatting Concerns:**

No concerns on paper formatting for this submission.

**Quality:**

3

**Strengths And Weaknesses:**

Strengths
1. The idea of patch-based autoregressive retrieval augmentation is well motivated, effectively addressing several limitations of image-level retrieval methods such as over-copying from references and stylistic bias.
2. The paper provides an in-depth exploration of the proposed paradigm through two well-designed implementations: (1) DAiD, a training-free method, and (2) FAiD, a parameter-efficient fine-tuning approach.
3. The proposed method achieves state-of-the-art performance, and the experimental results convincingly demonstrate the effectiveness of each component.

Weaknesses
1. Additional benchmark evaluations could further highlight the method’s general effectiveness.
- While the paper already evaluates the method on several standard benchmarks, including additional ones could strengthen the empirical support. Potential candidates include:
a) T2I-Benchmark [A], to assess improvements in image compositionality.
b) RareBench [B], to evaluate the benefits of patch-based retrieval in handling rare concepts.

2. Further ablation studies and analyses could enhance the paper’s completeness.
- A qualitative visualization of retrieved patches during generation would help assess retrieval quality and provide insight into how patch-based retrieval aids compositional generation.
- In addition to inference time, reporting GPU memory usage would offer a more comprehensive view of the computational overhead introduced by patch-based retrieval.
- Investigating the effect of the hyperparameter λ in DAiD (Eq. 4) would be valuable. An ablation study on this parameter could clarify the trade-off between model-predicted generation and retrieval-based guidance.

[A] T2i-compbench: A comprehensive benchmark for open-world compositional text-to-image generation, NeurIPS 2023

[B] Rare-to-Frequent: Unlocking Compositional Generation Power of Diffusion Models on Rare Concepts with LLM Guidance, ICLR 2025

---

> ### Author Rebuttal · Authors · 2025-07-31
>
> **Question:**  While the paper already evaluates the method on several standard benchmarks, including additional ones could strengthen the empirical support. Potential candidates include: a) T2I-Benchmark [A], to assess improvements in image compositionality. b) RareBench [B], to evaluate the benefits of patch-based retrieval in handling rare concepts.
>
> **Answer:** We thank the reviewer for these constructive suggestions. In response, we have conducted additional experiments on both RareBench and T2I-Bench, using the official prompts and evaluation protocols. The results are presented below.
> RareBench Results:
> | Models   | (single obj) Property | (single obj) Shape  | (single obj) Texture | (single obj) Action | (single obj) Complex | (multi obj) Concat | (multi obj) Relation | (multi obj) Complex |
> |----------|----------|--------|---------|--------|---------|--------|----------|---------|
> | RDM      | 68.00%   | 56.90% | 56.00%  | 44.00% | 64.50%  | 56.20% | 49.50%   | 39.50%  |
> | SDXL     | 60.00%   | 56.90% | 71.30%  | 47.50% | 58.10%  | 39.40% | 35.00%   | 47.50%  |
> | ImageRAG | 25.00%   | **86.50%** | **85.50%**  | 49.00% | 77.00%  | 57.50% | 36.00%   | 55.50%  |
> | Janus    | 53.50%   | 59.00% | 76.50%  | 53.50% | 78.00%  | 42.50% | 48.00%   | 50.50%  |
> | DAiD     | 64.00%   | 61.50% | 74.00%  | 54.00% | 78.50%  | 46.50% | 50.50%   | 54.50%  |
> | FAiD     | **74.00%**   | 62.00% | 78.50%  | **58.50%** | **79.00%**  | **67.00%** | **54.00%**   | **65.50%**  |
>
> T2I-Bench Results:
>
> | Models   | Color   | Shape    | Texture  | Numeracy |
> |----------|---------|----------|---------|----------|
> | RDM      | 0.666   | **0.555** | **0.538**  | 0.382    |
> | ImageRAG | 0.670   | 0.543    | 0.528   |  0.344    |
> | Janus    | 0.652   | 0.455    | 0.449   |  0.251    |
> | DAiD     | 0.630   | 0.459    | 0.464   |  0.245    |
> | FAiD     | **0.703** | 0.494    | 0.483   | **0.537** |
>
> All models in above two tabels are the same models used in Tabel 1 in our paper including DAiD and FAiD. We adopt the official prompts and evaluation code for both benchmarks.
>
> As observed on RareBench, our methods demonstrates substantial improvements over baselines, particularly for multi-object composition, indicating its effectiveness in generating rare or novel combinations not commonly found in the training data.
>
> On T2I-Bench, our methods also outperform the baseline Janus model by a considerable margin. While our model does not surpass the diffusion-based RDM model in all evaluation sub-categories, we respectfully note that this may be partly attributable to the relatively lower performance of the original Janus architecture. Notably, our method outperforms all baselines by a substantial margin in the "color" and "numeracy" categories. We did not report results for 2d-spatial and 3d-spatial categories, as all evaluated models achieved a score of 0 on these sub-tasks.
>
> ---
>
> **Question:**  A qualitative visualization of retrieved patches during generation would help assess retrieval quality and provide insight into how patch-based retrieval aids compositional generation.
>
> **Answer:**  We thank the reviewer for the excellent suggestion to visualize the retrieved patches during generation. In our visualization experiment, we take an image, randomly select a patch position, and retrieve the most relevant patch from the retrieval database conditioned on the surrounding patches. We then replace the selected patch with the retrieved one and observe that the retrieved patches integrate smoothly with their surrounding context.
>
> Due to conference policies, we are unable to submit additional PDF materials during the rebuttal phase. However, we will ensure that these visualizations are included in the revised version of the paper. Thank you again for this valuable suggestion.
>
> ---
>
> **Question:**  In addition to inference time, reporting GPU memory usage would offer a more comprehensive view of the computational overhead introduced by patch-based retrieval.
>
> **Answer:**  Here is the breakdown of inference memory usage for our models.
>
> Loading Janus Pro: 4717MB
>
> Performing a single image generation with Janus Pro: 5511MB
>
> Loading Janus Pro+FAiD: 5153MB
>
> Performing a single image generation with Janus Pro+FAiD: 5949MB
>
> Hosting Retrieval database: 77346MB
>
> As shown, incorporating FAiD modules results in only a modest increase in memory usage during both model loading and inference (approximately 400 MB). While the retrieval database itself requires additional memory, it is stored separately and can be efficiently managed. Overall, our retrieval approach introduces limited overhead relative to the improved performance it provides.
>
> ---
>
> **Question:**  Investigating the effect of the hyperparameter λ in DAiD (Eq. 4) would be valuable. An ablation study on this parameter could clarify the trade-off between model-predicted generation and retrieval-based guidance.
>
> **Answer:** We thank the reviewer for pointing out the importance of ablation on the hyperparameter λ in DAiD (Eq. 4). We apologize for not including this analysis in the main paper. As noted in Figure 7 in Appendix C.2 (Hyperparameter Optimization), we performed a hyperparameter search on λ to study its effect. In the revised version, we will ensure that this ablation study and the associated trade-offs are highlighted and discussed more prominently in the main text.

---

### Author Response · Authors · 2025-08-08

We appreciate the reviewers' recognition of our AR-RAG paper's strengths. All reviewers acknowledged the novelty and technical soundness of our approach, with Reviewer zZU3 highlighting how it effectively addresses limitations of image-level retrieval methods, Reviewer Dnu4 praising its creative multi-step retrieval strategy, Reviewer FaiV recognizing its rational use of patch-level information, and Reviewer L36Z noting its clever handling of inductive bias in image generation.

We've addressed all concerns raised during review. For Reviewer zZU3, we provided additional benchmark results on RareBench and T2I-Bench showing substantial improvements. For Reviewer Dnu4, we demonstrated through controlled experiments that our improvements stem from the retrieval mechanism rather than data similarity. For Reviewer FaiV, we clarified that our method has O(N) complexity with minimal overhead (0.22% additional time) and provided comparisons against SOTA diffusion models. For Reviewer L36Z, we conducted zero-shot generation experiments that confirmed our approach's effectiveness with reference images.

We're grateful to all reviewers for their constructive feedback, which has strengthened our paper. We particularly appreciate Reviewers Dnu4 and FaiV's acknowledgment that their concerns were resolved, leading to their reconsideration toward acceptance. We commit to incorporating all valuable insights into our final version.

---

### Decision · Program_Chairs · 2025-09-17

**Decision:**

Accept (poster)

**Comment:**

The paper proposes a RAG framework for AR based image generation i.e. RAIG for autoregressive generation.  The key contribution is to perform patch level retrieval at each iterative stage using previously generated patches.

The paper has received unanimous final recommendations to accept the paper (2x A, 2x BA), following clarifications made in the rebuttal largely around additional baselines and to address the common reviewer voice around complexity concerns.

zZUS had initial concerns about benchmarks and memory which were addressed. They are positive on the novelty and experimental sufficiency.

Dnu4 also had compute-cost related concerns that were addressed.  Additional baselines in the rebuttal converted the reviewer to a borderline accept.

FaiV much as Dnu4 had complexity and baselining concerns that were addressed.

L36Z was very positive and suggested some zero shot experiments that were mostly addressed in the rebuttal.

As all reviewers share a consensus view to accept, the AC recommendation is to accept.